# Conditional Independence Testing with Heteroskedastic Data and Applications to Causal Discovery

**Wiebke Günther**
German Aerospace Center
Institute of Data Science
07745 Jena, Germany
wiebke.guenther@dlr.de

**Urmi Ninad** *
Technische Universität Berlin
10623 Berlin, Germany
urmi.ninad@tu-berlin.de

**Jonas Wahl** *
Technische Universität Berlin
10623 Berlin, Germany
wahl@tu-berlin.de

**Jakob Runge**
German Aerospace Center
Institute of Data Science
07745 Jena, Germany
and
Technische Universität Berlin
10623 Berlin, Germany
jakob.runge@dlr.de

## Abstract

Conditional independence (CI) testing is frequently used in data analysis and machine learning for various scientific fields and it forms the basis of constraint-based causal discovery. Oftentimes, CI testing relies on strong, rather unrealistic assumptions. One of these assumptions is homoskedasticity, in other words, a constant conditional variance is assumed. We frame heteroskedasticity in a structural causal model framework and present an adaptation of the partial correlation CI test that works well in the presence of heteroskedastic noise, given that expert knowledge about the heteroskedastic relationships is available. Further, we provide theoretical consistency results for the proposed CI test which carry over to causal discovery under certain assumptions. Numerical causal discovery experiments demonstrate that the adapted partial correlation CI test outperforms the standard test in the presence of heteroskedasticity and is on par for the homoskedastic case. Finally, we discuss the general challenges and limits as to how expert knowledge about heteroskedasticity can be accounted for in causal discovery.

## 1 Introduction

Conditional independence (CI) testing is a frequently used step across a wide range of machine learning tasks for various scientific fields. It is also very challenging. Discovering causal relationships from purely observational data is an even more challenging task and an important topic in sciences where real experiments are infeasible, e.g. in climate research [Ebert-Uphoff and Deng, 2012, Runge et al., 2019a]. One can distinguish several frameworks that address this problem: score-based approaches [Chickering, 2002], restricted structural causal models [Peters et al., 2017], and constraint-based methods [Spirtes et al., 2000] that rely on CI testing. A typical representative is the PC algorithm [Spirtes and Glymour, 1991] which can be combined with any conditional independence

---

*equal contribution

36th Conference on Neural Information Processing Systems (NeurIPS 2022).

test and is thus adaptable to a wide range of data distributions. It utilizes the Faithfulness assumption to conclude that no causal link can exist between two variables $X$ and $Y$ if a CI test suggests that they are independent given a set $Z$.

When a linear additive noise model may be assumed, a popular CI test is the partial correlation test [Lawrance, 1976] which has the advantages of fast computation time and that the null distribution is known analytically. As a disadvantage rather strict and often unrealistic assumptions have to be satisfied. One of these assumptions is homoskedasticity, meaning that the variance of the error term is constant. When this assumption is violated, i.e., in the case of heteroskedasticity, the variance can, for instance, depend on the sampling index or the value of one or multiple influencing variables.

In regression analysis one distinguishes between *impure* and *pure* heteroskedasticity. Impure heteroskedasticity stems from an insufficient model specification, e.g. if unobserved variables or confounders are present. Another reason might be that the model fails to capture the full relationships of the variables, e.g. the underlying model might be linear with multiplicative noise rather than additive. On the other hand, pure heteroskedasticity is non-constant noise variance that is present despite a correct model.

The sources for heteroskedasticity in real data are manifold. For example, in environmental sciences precipitation in different areas might exhibit different variances that are unaccounted for by other variables in the system, i.e. location-scaled noise. Such a problem could be introduced by aggregating data of different catchments and not adding a variable that is well enough correlated with catchment location [Merz et al., 2021]. An example for sampling index-dependent heteroskedasticity in the time series case are seasonal effects that are present in many climate variables [Proietti, 2004]. Finally, an example where the noise variance of a variable is dependent on an observed cause is the distance to sea influencing the variability of temperature. This is a special case of state-dependent noise.

One assumption of the PC algorithm is causal sufficiency, meaning that there are no unobserved confounders, formally excluding impure heteroskedasticity. However, in practice this assumption is often violated which can lead to heteroskedastic noise.

If unaccounted heteroskedasticity is present in the data, the estimator of the ordinary least squares (OLS) regression slope parameter is still unbiased. However, the estimator of the covariance matrix of the parameter estimates can be biased and inconsistent under heteroskedasticity [Long and Ervin, 2000]. This can skew subsequent partial correlation significance tests and affect the link detection rate of a causal discovery method. Furthermore, heteroskedasticity might even lead to the detection of wrong links (false positives). Moreover, the Gauss-Markov theorem assumes homoskedasticity and, hence, with heteroskedastic data the OLS slope estimator is no longer guaranteed to be the most efficient linear unbiased estimator, which can further harm power of the CI test.

There are multiple ways to treat heteroskedasticity, either already during the modeling step by addressing possible model misspecification, by pre-processing the data, e.g. by applying a log-transform, or post-hoc using robust statistics for CI testing. Adapting the model, for example, by using CI tests allowing for multiplicative dependencies [Runge, 2018] might not always be feasible for limited sample sizes. Pre-processing in real-world problems also comes with drawbacks: The transformed variables are difficult to interpret and can introduce dependencies or spurious links.

In this work we propose an adapted weighted least-squares (WLS) partial correlation variant as a CI test for the PC algorithm that is able to deal with particular forms of heteroskedasticity. **Our contributions** are theoretical consistency results as well as numerical experiments demonstrating that this approach yields well-calibrated CI tests leading to controlled false positive rates and also improves upon detection power as compared to the standard partial correlation CI test. Our approach requires expert knowledge in that it needs to be known which of the variables the heteroskedasticity depends on, or if it depends on the sampling index.

## 2 Related Work

The effect of heteroskedasticity on the standard Pearson correlation test has been investigated, for example, by Wilcox and Muska [2001]. Remedies for heteroskedasticity also have been extensively studied. Hayes [2007] discuss and evaluate heteroskedasticity-consistent standard errors. Practical approaches to choose weights for WLS, consistency and asymptotic results have been obtained, e.g. by Neumann [1994], Fan and Yao [1998], Carroll [1982], Robinson [1987], Brown and Levine [2007].

Romano and Wolf [2017] propose a method that combines WLS with heteroskedasticity-consistent standard errors.

Recently, within the causal discovery framework of restricted structural causal models [Peters et al., 2017] several authors [Xu et al., 2022, Tagasovska et al., 2020] have relaxed the assumption of homoskedasticity. In these works the authors focus on identifying cause from effect only in the bivariate setting. Specifically, Xu et al. [2022] base their inference score on the log-likelihood of regression residuals and apply a binning-scheme on regions where variance is approximately constant.

More closely related to our work is the robust-PC method of Kalisch and Bühlmann [2008] who use an estimator in a recursive partial correlation formula to robustify the PC algorithm against outliers and otherwise contaminated data.

Non-constant conditional variance can also be regarded as a specific kind of distributional shift or context change. The effects of distributional shifts on causal discovery have been investigated, for instance, in Huang et al. [2020], Mooij et al. [2020] where the authors propose a framework that includes the environment into the structural causal model formulation using context variables.

## 3 Problem setting

### 3.1 Heteroskedasticity in causal models

In this work, we consider discovering causal relationships in linear models in the presence of non-constant error variance which is potentially dependent on the parents or on the sampling index. To translate this into a structural causal model (SCM), we represent heteroskedasticity as a scaling function of the noise variable. In this way, it can also be viewed as state-dependent or multiplicative noise.

Consider finitely many random variables $V = (X^1, \ldots, X^d)$ with joint distribution $\mathcal{P}_X$ over a domain $\mathcal{X} = \mathcal{X}_1 \times \ldots \times \mathcal{X}_d$. Then we are interested in $n$ samples from the following SCM with assignments

$$X_t^i := f_i(Pa(X_t^i)) + h_i(H(X_t^i)) \cdot N_i, \qquad i = 1, \ldots, d \tag{1}$$

where $f_i$ are linear functions, $t \in \mathcal{T}$ stands for the sample index, and we have the heteroskedasticity functions $h_i : \mathcal{X} \times \mathcal{T} \to \mathbf{R}_{\geq 0}$. The noise variables $N_i$ are assumed independent standard Gaussian distributions. The parent set of the variable $X_t^i$ is denoted by $Pa(X_t^i)$, and $H(X_t^i) \subset Pa(X_t^i) \cup \{t\}$ which can also be the empty set. Furthermore, we make the restriction that the causal relationships are stable over time, i.e. the parent sets $Pa(X_t^i)$ as well as the functions $f_i$ are not time-dependent.

For $h_i \equiv const.$ the homoskedastic version of this SCM is obtained, which simply is a linear SCM with Gaussian noise. To learn such a causal model from data, a well-known and easy to implement method is to use a constraint-based causal discovery method with CI tests based on partial correlation.

To recap the partial correlation test and to illustrate our adaptations, consider the following simple example model for a time-series, or otherwise indexed, SCM (1). The general multivariate case is discussed in section 4.2.

$$\begin{aligned} X_t &= aZ_t + cE_t + h_X(Z_t, t) \cdot N_X & Z_t &= N_Z \\ Y_t &= bZ_t + cE_t + h_Y(Z_t, t) \cdot N_Y & E_t &= N_E \end{aligned} \tag{2}$$

for some constants $a$, $b$ and $c$, and standard normal independent noise terms $N_X$, $N_Y$, $N_Z$, and $N_E$. The constant $c$ is set to zero for (conditionally) independent $X$ and $Y$. Note that here we are only interested in discovering the relationship between $X$ and $Y$ and not in learning the whole causal graph.

An intuitive way to define and understand partial correlation is in terms of the correlation between residuals. The partial correlation between $X$ and $Y$ given a controlling variable $Z$, or a set thereof, is the correlation between the residuals $r_X$ and $r_Y$ resulting from the linear regression of $X$ on $Z$ and of $Y$ on $Z$, respectively. The linear regression that is used in the standard variant of the partial correlation test is ordinary least squares (OLS) regression, thus we refer to this CI test as *ParCorr-OLS*. The Pearson correlation coefficient $\rho$ is then estimated by $\widehat{\rho}(X, Y|Z) = \frac{\widehat{Cov(r_X, r_Y)}}{\sqrt{\widehat{Var(r_X)Var(r_Y)}}}$. We use the hat operator to indicate estimators. For testing the null hypothesis $H_0 : \rho = 0$ versus the alternative

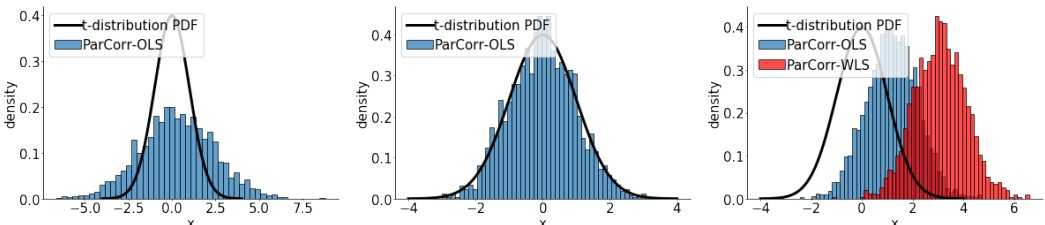

Figure 1: Data generated with SCM (2). (Left) Null distribution when $X$ and $Y$ are affected by the same kind of heteroskedasticity. (Middle) Null distribution when only $X$ is affected by heteroskedasticity. (Right) Alternative distribution (assuming dependence with $c \neq 0$) when only $X$ is affected by heteroskedasticity. In all plots the heteroskedasticity is a linear function of the value of $Z$ and the solid black line depicts the $t$-distribution under the null hypothesis.

$H_1 : \rho \neq 0$, we use the studentized version of the partial correlation as a test statistic $T(\hat{\rho}) = \frac{\hat{\rho}\sqrt{n-2}}{\sqrt{1-\hat{\rho}^2}}$.
This statistic is $t(n - 2 - k)$-distributed, where $k$ is the number of variables we are conditioning on, and $n$ is the sample size.

However, for the partial correlation test to be correct, the assumptions of OLS, in particular homoskedasticity, have to be fulfilled. In the next section, we investigate what happens if this assumption is violated.

## 3.2   Effects of heteroskedasticity on partial correlation

Under heteroskedasticity, the estimator of slope regression parameters in OLS regression is still unbiased. However, the estimator of the covariance matrix of the parameter estimates can be biased and inconsistent under heteroskedasticity [Wooldridge, 2009, p.268]. This also affects the subsequent residual-based correlation test in ways that we summarize below. Proofs for and further discussions of the following statements are provided in the Supplement A.1.

**Effect 1.** *Under the null hypothesis, the studentized Pearson correlation coefficient is $t(n - 2 - k)$ distributed if $X$ is independent of the variables inducing heteroskedasticity in $Y$.*

This means that if only one node is compromised by heteroskedasticity, or, more generally, the heteroskedasticity in $X$ is independent of that in $Y$, the type-I error rate of the t-test will not be affected. Please refer to the middle plot in figure 1 for a visualization of the associated null distribution in comparison to the analytical one used by the partial correlation test. On the other hand, in the left plot the case where both $X$ and $Y$ are affected by the same kind of heteroskedasticity is shown, leading to a different null distribution.

**Effect 2.** *If $X$ and $Y$ are dependent and at least one is affected by heteroskedasticity, the detection power of the t-test might be degraded.*

In particular, if $X$ and $Y$ are dependent, $X$ is affected by linear heteroskedasticity, the mean of the $t$-distribution is closer to zero compared to the distribution of weighted least squares based studentized partial correlation coefficient, which is introduced in the next section and essentially transforms the data to be homoskedastic, for fixed sample size. See the right plot in figure 1. This effect is an immediate consequence of the reduced efficiency of the OLS slope estimate. Intuitively, heteroskedasticity masks the relationship between $X$ and $Y$.

## 4   Weighted least squares partial correlation test and causal discovery

We have seen that the standard partial correlation test is sensitive to heteroskedastic noise since it is based on an OLS regression step. Therefore, we propose to replace the OLS regression by the weighted least squares (WLS) approach which is known to be able to handle non constant error variance. We will refer to the resulting CI test as *ParCorr-WLS*.

## 4.1 Weighted least squares partial correlation test

The idea of WLS is to perform a re-weighting of each data point depending on how far it is from the true regression line. It is reasonable to assume data points where the error has low variance to be more informative than those with high error variance. Therefore, ideally the weights are chosen as the inverse variance of the associated error.

To formalize this idea, consider the linear model $y = X\beta + \varepsilon$ with $\mathrm{E}[\varepsilon \mid X] = 0$, $\mathrm{Cov}[\varepsilon \mid X] = \mathrm{diag}(\sigma_i^2)_{i=1,\ldots,n}$ for $n$ observations. The variance of the error term $\varepsilon_i$ is denoted by $\sigma_i^2$. Note, as opposed to the assumptions of OLS, the entries of the conditional variance matrix are allowed to differ from each other. Denote the weight matrix by $W := \mathrm{diag}(\frac{1}{\sigma_i^2})_{i=1,\ldots,n}$. The WLS method estimates $\beta$ by solving the adjusted optimization problem

$$\hat{\beta} = \underset{b}{\mathrm{argmin}}(y - Xb)^{\mathsf{T}} W (y - Xb).$$

This objective is quadratic, thus we can write down the solution in closed form

$$\hat{\beta} = \left(X^{\mathsf{T}} W X\right)^{-1} X^{\mathsf{T}} W y.$$

If the true weights are known, WLS is equivalent to applying OLS to a linearly transformed, homoskedastic version of the data as the weights have the effect of standardizing the scale of the errors. Thus, the following lemma holds. Proofs for these statement can be found, for instance, in Greene [2003].

**Lemma 1.** *If the weights are chosen as the reciprocal of the error variance per sample, the WLS estimator is consistent, efficient, and asymptotically normal, as well as BLUE.*

Moreover, note that the weighted residuals $W(y - X\hat{\beta})$ are homoskedastic.

The goal now is to approximate the conditional variance function of both $X$ and $Y$ in SCM (2). Since it works analogously for both cases, we illustrate the approach for $X_t = aZ + s(Z, t) \cdot N_X$, i.e. we approximate $\sigma^2(z, t) = Var(X_t|Z = z)$.

For that, we use a residual-based non-parametric estimator for the conditional variance, similar to the approach of Robinson [1987]. Motivated by the identity $Var(X|Z) = \mathbb{E}[(X - \mathbb{E}[X|Z])^2|Z]$, and noting that this is the regression of $(X - aZ)^2$ on $Z$, the first step is using OLS regression to obtain the squared residuals $(X - \hat{a}Z)^2$. Afterwards, we use a non-parametric regression method to regress these residuals on $Z$ and thereby predict the conditional mean by using a linear combination of the $k$ residuals closest in $Z$ value. For sampling index-dependent heteroskedasticity this turns into a windowing approach, which essentially smoothes the squared residuals.

Algorithm 1 details the proposed partial correlation CI test based on the feasible WLS approach that employs our weight approximation method. For the test to perform well, it is crucial to know the type of heteroskedasticity, more precisely, which of the predictors the variance depends on. In practice, this kind of expert knowledge could be obtained by performing a test for heteroskedasticity, e.g. as suggested in Wooldridge [2009, p.277] or by investigating plots of the residuals. Here we make the limiting assumption that the heteroskedasticity only depends on one of the predictors or on the sampling index. Further extensions are considered in the section 6.

Now, we formulate our main assumption under which it is possible to obtain a consistency result for the WLS method that uses our weight approximation method.

**Assumption 1** (Heteroskedastic relationships). *For each node $X^i$, $i = 1, \ldots, d$ in SCM (1), the skedasticity or noise scaling function $h_i$ only depends on one of the predictors $H$ or on the sampling index $t$, i.e. its domain is one dimensional, or it is constant. Furthermore, it is known what it depends on.*

We also have to impose a rather technical assumption on the functions $h_i$.

**Assumption 2** (Weight approximation). *Assumptions (3.3) - (3.5) from Robinson [1987].*

The proof of the following lemma can be found in Robinson [1987].

**Lemma 2.** *Under assumptions 1 and 2, the WLS estimator that uses the reciprocal of the approximated variance as weights is consistent and attains the correct covariance matrix asymptotically.*

**Assumption 3** (Technical assumptions)**.** *See Supplement A.2.*

We now state the first main theorem.

**Theorem 1.** *Under the assumptions 1 - 3 ParCorr-WLS CI test with estimated weights (Algorithm 1) is consistent for testing the conditional independence between two potentially heteroskedastic variables $X$ and $Y$ conditioned on a set of variables $Z$.*

---

**Algorithm 1:** ParCorr-WLS

---

**Data:** Expert knowledge $\mathcal{E}$ as map with keys $X$ and $Y$ and values in {false, 'sampling index', 'heteroskedastic parent $H$'}, where $H \in V$, observational data with sample size $n$ for nodes $X, Y, Z_1, \ldots, Z_k, H$, window length $\lambda$, significance level $\alpha$

**Result:** boolean indicating whether there is a (conditional) dependence between $X$ and $Y$ given $Z_1, \ldots, Z_k$

$resid := []$;

**for** *node in {X, Y}* **do**

    **if** $k == 0$ **then**

        | $\tilde{r} = node$

    **else**

        | obtain residuals $\tilde{r} = (\tilde{r}_1, \ldots, \tilde{r}_n)$ by regressing *node* on $Z_1, \ldots, Z_k$ using OLS;

    **end**

    **if** $\mathcal{E}[node] == $ *'false'* **then**

        | append $\tilde{r}$ to $resid$;

    **else if** $\mathcal{E}[node] == $ *'sampling index'* **then**

        compute weights $w_i = (\frac{1}{\lambda} \sum_{j=\max(1, i-\frac{\lambda}{2})}^{\min(n, i+\frac{\lambda}{2})} \tilde{r}_j^2)^{-1}$, for $i = 1, \ldots, n$;

        obtain residuals $r$ by regressing *node* on $Z_1, \ldots, Z_k$ using WLS with the weights $w$;

        append $w \cdot r$ to $resid$;

    **else**

        sort $\tilde{r}$ such that their corresponding values of $H$ increase;

        compute weights $w_i = (\frac{1}{\lambda} \sum_{j=\max(1, i-\frac{\lambda}{2})}^{\min(n, i+\frac{\lambda}{2})} \tilde{r}_j^2)^{-1}$, for $i = 1, \ldots, n$;

        revert the sorting in the indices of $w$;

        obtain residuals $r$ by regressing *node* on $Z_1, \ldots, Z_k$ using WLS with the weights $w$;

        append $w \cdot r$ to $resid$;

**end**

calculate studentized Pearson correlation $t$ between $r_X = resid[0]$ and $r_Y = resid[1]$;

perform $t$-test with (two-sided) significance level $\alpha$, i.e. reject if $|t| > t(1 - \frac{\alpha}{2}, n - 2 - k)$

---

## 4.2 Extension to the PC algorithm

A well-known and widely used algorithm for discovering causal relationships in terms of the completed partially directed acyclic graph (CPDAG) from observational data is the PC algorithm as introduced in Spirtes and Glymour [1991]. It consists of two phases: The first one is concerned with learning the skeleton of adjacencies based on iterative CI testing. Subsequently, a set of rules is applied to determine the orientation of the found links.

To ensure consistency of this method, the following assumptions have to be fulfilled. Details can be found in Spirtes et al. [2000]. Let $G = (V, E)$ be a graph consisting of a set of vertices $V$ and a set of edges $E \subset V \times V$. Let $\mathcal{P}$ denote the probability distribution of $V$.

**Assumption 4** (PC algorithm)**.** *The Causal Markov condition, Causal Faithfulness, and Causal Sufficiency are fulfilled for the SCM* (1) *with graph $G$.*

Under these assumptions and if the utilized conditional independence test is consistent, it can be shown that the PC algorithm converges in probability to the correct causal structure, i.e.

$$\lim_{n \to \infty} \mathcal{P}(\hat{G}_n \neq G) = 0,$$

where $G$ denotes the ground truth CPDAG and $\hat{G}_n$ is the finite sample output of the PC algorithm. See Kalisch and Bühlmann [2007] for a proof.

In the following, we need to further restrict Assumption 1 to prove consistency of the PC algorithm under heteroskedasticity.

**Assumption 5** (Heteroskedastic relationships regarding PC algorithm)**.** *Assumption 1 is further limited to the case where there is only sampling index dependent heteroskedasticity or homoskedasticity.*

Given Assumption 5, we can apply our proposed method ParCorr-WLS in every CI test of the PC algorithm. Using Lemma 1, we thus can establish consistency of the PC algorithm with ParCorr-WLS if the true weights are known. Lemma 2 yields the same result for ParCorr-WLS with estimated weights.

**Theorem 2.** *Under Assumptions 2,3,4,5 the output of the PC algorithm, with ParCorr-WLS as a CI test, converges in probability to the correct causal graph.*

Note that we prove consistency of the PC algorithm with ParCorr-WLS only in the case of sampling index dependent heteroskedasticity. We discuss the challenges to extending this to more general types of heteroskedasticity in section 6.

## 5 Experiments

In the following, we conduct experiments evaluating our proposed CI test separately and in conjunction with the PC algorithm. Throughout the experiments, heteroskedasticity strength refers to the parameter $s$ in the scaling functions $h$ of linear and periodic type given by

$$
\begin{aligned}
h(x) &= 1 + se^T x \cdot \mathbb{1}_{x \geq 0} \quad \text{(linear)} \\
h(x) &= 1 + se^T \sin(x) + s \quad \text{(periodic)} .
\end{aligned}
\tag{3}
$$

In other words, in case of linear heteroskedasticity strength is the slope of the variance function, and for periodic heteroskedasticity it refers to the amplitude of the variance function.

### 5.1 Conditional independence testing

We generate the data from the SCM (2) where we consider various types of heteroskedasticity, i.e. functions $h$, namely linear and periodic as given in Eq. (3). Plots of the simulated data are provided in the Supplement. Each of the types can either be $Z$- or sampling index-dependent. A visualization of the considered heteroskedasticity-types can be found in the Supplement A.5.

We use the Kolmogorov-Smirnov (KS) statistic to quantify how uniform the distribution of p-values is, and therefore as a metric for type-I errors, as in Runge [2018]. Type-II errors are measured by the area under the power curve (AUPC). The metrics were evaluated with a sample size of 500 from 100 realizations of the SCM. Error bars indicate the bootstrapped standard errors.

Figure 2 shows that ParCorr-WLS is well calibrated in the presence of heteroskedasticity, regardless if it affects only one or both of the variables $X$ and $Y$. On the other hand, the ParCorr-OLS test becomes ill-calibrated in the case of heteroskedastic $X$ and $Y$ as expected by Effect 1. In particular, this means that if we can choose the weights reasonably well, we are able to overcome multiplicative confounding with our proposed CI test.

Regarding power as measured by AUPC, we observe a rather rapid decrease for ParCorr-OLS as the heteroskedasticity increases (compare to Effect 2). Our proposed method ParCorr-WLS has higher power in the heteroskedastic scenario, and even for homoskedastic noise (heteroskedasticity strength equal zero) the power is comparable to that of ParCorr-OLS. The drop in power for ParCorr-WLS can be explained by an overall increase in noisiness of the data as heteroskedasticity strength is growing. Refer to the Supplement A.5 for a plot of AUPC of ParCorr-OLS and ParCorr-WLS on data with homoskedastic but increasing noise.

Similar effects are present for all considered types of heteroskedasticity (see Supplement). The results remain very similar if we do not use the ground truth weights but estimate them using the window approach with a reasonable window length as detailed in Section 4.1.

### 5.2 Causal discovery

To test our proposed CI test in a more realistic setting, we apply it within the PC algorithm to recover the causal graph from simulated observational data. We build upon the PC-stable algorithm

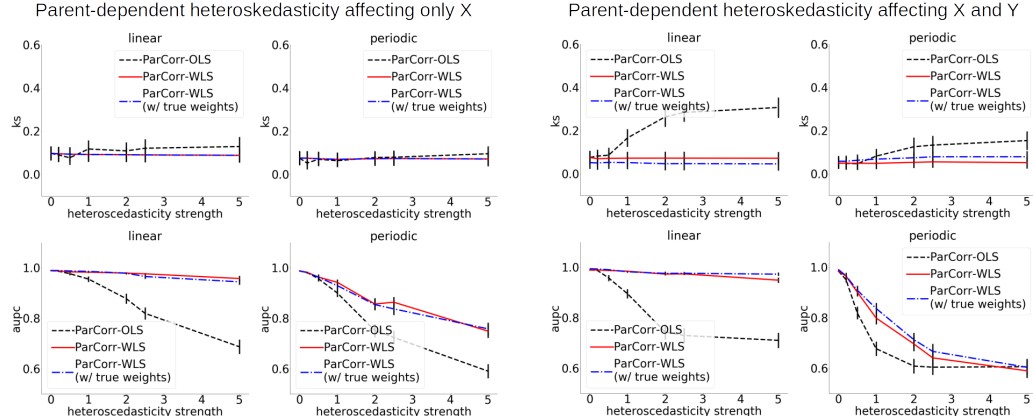

Figure 2: Performance of partial correlation CI tests for dependence ($c = 0.5$) and conditional independence ($c = 0$) between $X$ and $Y$ given $Z$. In all plots the heteroskedasticity is a function of the confounder $Z$ and only affects $X$ (left two columns) or both $X$ and $Y$ (right two columns). Shown are KS (top row) and AUPC (bottom row) for different strengths of heteroskedasticity for linear (left), periodic (right) noise scaling functions. The ground truth weights are used for WLS, or the weights are estimated using the window approach with window length $10$ as detailed in section 4.1. A sample size of $500$ is used and the experiments are repeated $100$ times.

implementation within the Tigramite software package [Runge et al., 2019b] which is published under the GNU General Public License.

The data for these experiments was generated with SCM (1) in the following way. Given a random ground truth graph, we fix a percentage of heteroskedasticity-affected nodes which are then selected uniformly at random. Throughout the experiments this percentage is set to $0.3$ to reduce the chance of parent and child being affected by the same kind of heteroskedasticity. For these affected nodes, we choose as a heteroskedasticity type linear or periodic with equal probability (Eq. (3)) and let the noise variance $h$ either depend on one randomly selected parent or the sampling index. We also set a fixed strength $s$ per experiment and investigated the effect of increasing the strength on the performance of our method compared to the PC algorithm with ParCorr-OLS. All linear dependencies have a coefficient $c = 0.5$. We tested for various small to medium sized causal graphs (see also Supplement).

In these experiments, we estimate the weights based on the expert knowledge as required by Assumption 1, namely which node is affected by heteroskedasticity, and whether the noise variance depends on the sampling index or another node. In the case of the heteroskedasticity depending on another node, we also need to know which node it is. We also compare with the PC algorithm which uses ParCorr-WLS based on the ground truth weights for all direct heteroskedastic relations. Note, however, that this does not take care of indirect heteroskedasticity due to heteroskedastic parents that are not part of the conditioning set. Note that this challenging setting is outside of the stricter Assumption 5 for which the PC algorithm is consistent.

Figure 3 shows, similar to the experiments of the previous section, that the PC algorithm with ParCorr-WLS continues to have a rather small false positive rate (FPR) even though the heteroskedasticity strength increases. In contrast, ParCorr-OLS shows an increase in FPR as the heteroskedasticity strength increases. Additionally, even in this rather complicated setting, we see that the true positive rate (TPR) can be improved by using our method. The PC algorithm with ParCorr-WLS has an average runtime of $0.29$ seconds on homoskedastic data compared to $0.14$ seconds for ParCorr-OLS evaluated on AMD 7763.

## 6 Discussion and Outlook

In this work we relaxed the common assumption of a constant variance by explicitly allowing for heteroskedasticity as a multiplicative scaling of noise in an otherwise linear SCM. Our proposed partial correlation test based on weighted least squares regression is a linear, computationally fast and

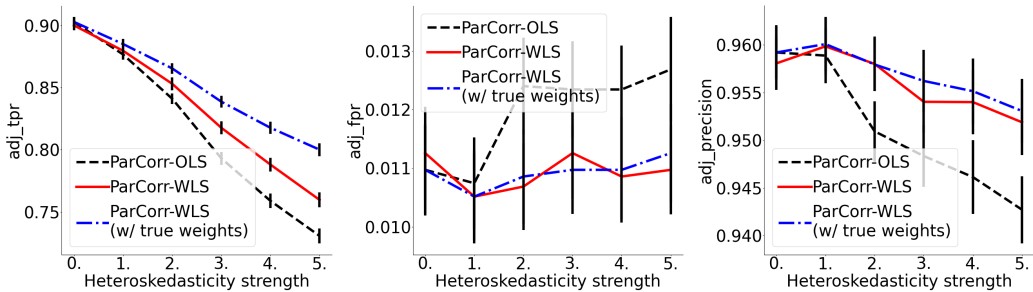

Figure 3: Results for the PC algorithm with the standard ParCorr-OLS CI test compared to that with the proposed ParCorr-WLS test. Shown are adjacency TPR (left) and FPR (middle), as well as the adjacency precision (right) for increasing strengths of heteroskedasticity. The graph has 10 nodes and 10 edges, a sample size of 500 is used. The significance level $\alpha$ is set to 0.05. The experiment is repeated 500 times. Errorbars show standard errors. Estimated weights with a window length of 5 or ground truth weights are used for WLS.

easy to implement method and constitutes a useful standalone method for various data science tasks, but our focus here is on its use in causal discovery. The main **strengths** of our approach are that it is able to produce more reliable results both as a CI test and in causal discovery in the presence of heteroskedasticity than the standard partial correlation variant. More reliable results here refers to controlled false positives as well as higher detection power. Furthermore, the suggested adaptations do not compromise calibratedness and power of the CI test on homoskedastic data.

The main **weakness** of our method is that the CI test requires substantial expert knowledge in Assumption 1 and for the consistency of the PC algorithm the even stricter Assumption 5. Assumption 1 requires that the heteroskedasticity only depends on one of the predictors or on the sampling index, i.e. its domain is one dimensional. Furthermore, one needs to know which of these types is the case. Assumption 5 only allows for sampling-index heteroskedasticity. The reason for this is that there can be indirect heteroskedasticity at the node $X$ or $Y$ that is not induced by a parent of $X$ or $Y$, but by some other ancestor. This heteroskedasticity then propagates through the causal graph and essentially makes $X$ or $Y$ heteroskedastic whenever this path is not blocked by the conditioning set. In this case, the expert knowledge does not tell us about this heteroskedasticity and we would not be able to apply ParCorr-WLS to remove it. See also Section A.4 in the Supplement for a detailed treatment of cases in which the CI tests within the PC algorithm are consistent under more general heteroskedasticity forms. In the following, we discuss alternatives and further avenues of research.

**In future work** one may alter the required expert knowledge and weight approximation scheme to overcome the issues induced by indirect heteroskedasticity. Here, a possible remedy could be an iterative approach using information about causal relationships from earlier steps. An alternative would be to use the CMIknn CI test [Runge, 2018] that is able to fully treat heteroskedastic multiplicative confounding. However, this increased generality comes at the price of reduced detection power, also see Figure 10 in the Supplement.

Another open question is how to further improve the weight approximation method. For instance, by iteratively repeating the regression and smoothing steps. Another important consideration is the choice of the window length. Linear properties of the variance function allow us to use a larger window length. However, if the variance function shows a high variability, crucial information might be lost if the chosen window length is too large. Model selection criteria might be employed to alleviate this problem.

Furthermore, one can consider extensions of ParCorr-WLS to multiple heteroskedastic influencing factors, e.g., Spokoiny [2002] extend the residual-based conditional variance function estimation to dimensions larger than one. Another multidimensional method based on differences is discussed in Cai et al. [2009].

It would also be interesting to explore using generalized least squares to be able to account for additional correlation of the residuals. Potentially, the weighted least squares partial correlation

coefficient could also be combined with a permutation-based test to extend the method to problems with non-Gaussian noise.

**Concluding**, our proposed ParCorr-WLS CI test makes constraint-based causal discovery methods better applicable to real world problems where the assumption of homoskedasticity is violated, such as in climate research or neuroscience. Ethically, we believe that our rather fundamental work has a low potential for misuse.

## Acknowledgments and Disclosure of Funding

WG was supported by the Helmholtz AI project *CausalFlood*. UN and JW were supported by grant no. 948112 *Causal Earth* of the European Research Council (ERC). This work used resources of the Deutsches Klimarechenzentrum (DKRZ) granted by its Scientific Steering Committee (WLA) under project ID bd1083. We thank the anonymous reviewers for their helpful comments.

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
