# A Supplement

Here we provide proofs of the statements made in the main text as well as further figures of numerical experiments and a more detailed discussion of heteroskedasticity effects regarding causal discovery.

## A.1 Proof of Effects 1 and 2

### A.1.1 Proof of Effect 1:

*Proof.* Let $(X_i, Y_i)_{i=1,\dots,n}$ be an independent sample with Pearson correlation coefficient $\rho$, and we assume the linear model $Y_i = X_i\beta + h(Z_i)\epsilon_i$, where $Z_i$ and $\epsilon_i$ are independent and standard normal, and $h$ is the noise scaling function.

Note that w.l.o.g. we assume only $Y$ to be heteroskedastic w.r.t. $Z$.

Testing whether the Pearson correlation between $X$ and $Y$ is zero is equivalent to testing whether the slope parameter $\beta$ is equal to zero.

Under the null hypothesis, calculating $\mathrm{Var}(h(Z_i)\epsilon_i|X_i)$ gives

$$\mathrm{Var}(h(Z_i)\epsilon_i|X_i) = \mathbb{E}[h^2(Z_i)\epsilon_i^2|X_i] = \mathbb{E}[h^2(Z_i)\epsilon_i^2] = \mathbb{E}[h^2(Z_i)] = \mathbb{E}[h^2(Z_1)]$$

using the independence of $Z_i$ and $X_i$, the fact that $\epsilon_i$ is standard normal, and that the $Z_i$ are identically distributed. Therefore, this is a homoskedastic problem.

$\square$

### A.1.2 Discussion of Effect 2:

We start by discussing the homoskedastic case to see where non-constant variance of noise leads to problems within the t-test. Let $(X_i, Y_i)_{i=1,\dots,n}$ be an independent sample from a bivariate normal distribution with Pearson correlation coefficient $\rho$, and we assume the linear model $Y_i = X_i\beta_0 + \epsilon_i$, where $\epsilon_i$ is standard normal, then

$$\beta_0 = \rho\sqrt{\frac{\mathrm{Var}(Y)}{\mathrm{Var}(X)}}.$$

This also is true for the finite sample estimators of these entities, since

$$\hat{\rho} = \frac{\sum X_i Y_i}{\sqrt{\sum X_i^2 \sum Y_i^2}}$$

and

$$\hat{\beta} = \beta_0 + \frac{\sum X_i \varepsilon_i}{\sum X_i^2} = \beta_0 + \frac{\sum X_i(Y_i - X_i\beta_0)}{\sum X_i^2} = \frac{\sum X_i Y_i}{\sum X_i^2} = \hat{\rho}\sqrt{\frac{\sum Y_i^2}{\sum X_i^2}}.$$

Furthermore, the following relationship holds for the estimator of the studentized correlation coefficient and the estimator of the slope parameter $\beta_0$

$$\hat{\rho}\frac{\sqrt{n-2}}{\sqrt{1-\hat{\rho}^2}} = \hat{\beta}\sqrt{\frac{\sum X_i^2}{\frac{1}{n}\sum(Y_i - \hat{\beta}X_i)^2}}. \tag{4}$$

For homoskedastic noise the second factor is an estimator of the standard error of $\hat{\beta}$, which we derive by using the mean of the squared residual as an estimator for the error variance. We know that

$$\hat{\beta} = \frac{\sum X_i Y_i}{\sum X_i^2} = \frac{\sum X_i(\beta_0 X_i + \epsilon_i)}{\sum X_i^2} = \beta_0 + \frac{\sum X_i \epsilon_i}{\sum X_i^2}$$

and thus

$$\mathrm{Var}(\hat{\beta}) = \mathrm{Var}\Big(\frac{\sum X_i \epsilon_i}{\sum X_i^2}\Big). \tag{5}$$

If we assume the design to be fixed and $\mathrm{Var}(\epsilon_i) = \sigma^2$ to be constant, this simplifies to

$$\mathrm{Var}(\hat{\beta}) = \frac{\sum X_i^2 \mathrm{Var}(\epsilon_i)}{(\sum X_i^2)^2} = \frac{\sigma^2 \sum X_i^2}{(\sum X_i^2)^2}.$$

Therefore we use the following estimator

$$\widehat{\mathrm{Var}(\hat{\beta})} = \frac{\frac{1}{n}\sum (Y_i - \hat{\beta}X_i)^2}{\sum X_i^2}. \tag{6}$$

We now assume that $Y$ is heteroskedastic, i.e. the model takes the form

$$Y_i = X_i\beta_0 + h(Z_i)\epsilon_i,$$

We need to show

$$\mathbb{E}\left[\frac{\hat{\beta}^{\mathrm{OLS}}}{\sqrt{\widehat{\mathrm{Var}(\hat{\beta}^{\mathrm{OLS}})}}}\right] \le \mathbb{E}\left[\frac{\hat{\beta}^{\mathrm{WLS}}}{\sqrt{\widehat{\mathrm{Var}(\hat{\beta}^{\mathrm{WLS}})}}}\right],$$

where $\hat{\beta}^{\mathrm{OLS}}$ is the OLS estimator for the slope parameter and $\hat{\beta}^{\mathrm{WLS}}$ is its WLS version.

Since OLS and WLS generally return similar values for the slope estimator (especially for large $n$) and both are unbiased estimators for $\beta_0$, we focus on the denominator.

We get $\mathbb{E}\left[\frac{\beta_0}{\sqrt{\widehat{\mathrm{Var}(\hat{\beta}^{\mathrm{OLS}})}}}\right] \le \mathbb{E}\left[\frac{\beta_0}{\sqrt{\widehat{\mathrm{Var}(\hat{\beta}^{\mathrm{WLS}})}}}\right]$ from the reduced efficiency of the OLS regression on heteroskedastic data, i.e. from

$$\mathrm{Var}(\hat{\beta}^{\mathrm{OLS}}) \ge \mathrm{Var}(\hat{\beta}^{\mathrm{WLS}}),$$

if we assume that the estimators (6) for $\mathrm{Var}(\hat{\beta}^{\mathrm{OLS}})$ and $\mathrm{Var}(\hat{\beta}^{\mathrm{WLS}})$ are unbiased. The unbiasedness is clear for the WLS-case, as well as for the case where the heteroskedasticity in $X$ is independent from the one in $Y$, see Effect 1. In the case of heteroskedasticity in $Y$ that is dependent on $X$, the estimator might be biased. If it is over-estimating $\mathrm{Var}(\beta)$, the power is also reduced. The situation of under-estimating the variance can, in fact, quasi-increase power but comes with the major drawback of increased probability of type I error.

## A.2 Proof of Theorem 1

### A.2.1 Assumptions

Consider random variables $X = (X_1, \ldots, X_n)^T, Y = (Y_1, \ldots, Y_n)^T$ and $Z^k = (Z_1^k, \ldots, Z_n^k)^T$ for $k = 1, \ldots, d$, where $n$ is the number of samples. Denote the design matrix of the regressor variable by $Z = (Z^1, \ldots, Z^d) \in \mathbb{R}^{n \times d}$. We assume that all variables are centered and generated according to the model

$$X = Z \cdot \alpha + \varepsilon^X$$
$$Y = Z \cdot \beta + \varepsilon^Y$$

with error variables $\varepsilon^X = (\varepsilon_1^X, \ldots, \varepsilon_n^X)^T, \varepsilon^Y = (\varepsilon_1^Y, \ldots, \varepsilon_n^Y)^T$. We make the following assumptions:

(A1) The error vectors $\varepsilon^X, \varepsilon^Y$ have zero mean and diagonal covariance matrices $\Sigma_X = \mathrm{diag}(\sigma_{X,1}^2, \ldots, \sigma_{X,n}^2), \Sigma_Y = \mathrm{diag}(\sigma_{Y,1}^2, \ldots, \sigma_{Y,n}^2)$. Similarly, for any $k$, the regressor variables $Z^k = (Z_1^k, \ldots, Z_n^k)^T$ have diagonal covariance matrices.

(A2) The matrix $Z^T Z$ is almost surely positive definite, of full rank $d$, and we have that

$$\lim_{n \to \infty} \frac{Z^T Z}{n} = Q$$

converges entrywise in probability to a matrix $Q$. Note that the $k, \ell$-entry of $Z^T Z$ is $\sum_{i=1}^n Z_i^k Z_i^\ell$.

(A3)  We have well defined asymptotic average variances

$$\lim_{n\to\infty} \frac{1}{n} \sum_{i=1}^{n} \sigma_{X,i}^2 = \sigma_X^2$$

$$\lim_{n\to\infty} \frac{1}{n} \sum_{i=1}^{n} \sigma_{Y,i}^2 = \sigma_Y^2$$

and the limits

$$\lim_{n\to\infty} \frac{1}{n} \sum_{i=1}^{n} (Z_i^k)^2 \sigma_{X,i}^2, \qquad \lim_{n\to\infty} \frac{1}{n} \sum_{i=1}^{n} (Z_i^k)^2 \sigma_{Y,i}^2 \qquad \text{for } k = 1, \ldots, d$$

exist in probability.

(A4)  The variances of each variable are bounded by a constant independent of the sampling index $i$.

(A5)  For parent-dependent heteroskedasticity, let $H$ be the heteroskedasticity-inducing parent. Denote by $\mathcal{S}_i$ the set of $\lambda$ nearest neighbours of $X_i$ (or $Y_i$ respectively) in $H$-value. In the case of sampling index dependent heteroskedasticity, $\mathcal{S}_i$ denotes the set of nearest neighbours in sampling index value.
We assume

$$\lambda \to \infty, \quad \frac{\lambda}{n} \to 0 \quad \text{as} \quad n \to \infty.$$

Furthermore, the limit of the variances averaged over the nearest neighbours exists and converges to the right value, i.e.

$$\lim_{n\to\infty} \frac{1}{\lambda} \sum_{r_j \in \mathcal{S}_i} \sigma_{X,j}^2 = \sigma_{X,i}^2, \qquad \lim_{n\to\infty} \frac{1}{\lambda} \sum_{r_j \in \mathcal{S}_i} \sigma_{Y,j}^2 = \sigma_{Y,i}^2.$$

### A.2.2  Proof

We will now sketch the proof of consistency of the estimator for the partial correlation $\rho_{X,Y|Z}$ in the case where we regress on only one variable, i.e. $d = 1$. The result can however easily be generalized to higher dimensions although the computations become slightly more involved. Recall that in the first step of ParCorr-WLS, we compute the residuals

$$r_i^X = (\alpha - \hat\alpha)Z_i + \varepsilon_i^X, \quad r_i^Y = (\beta - \hat\beta)Z_i + \varepsilon_i^Y,$$

where $\hat\alpha, \hat\beta$ are OLS-estimates of $\alpha, \beta$. More precisely, recall that $\hat\alpha$ is given by the formula

$$\hat\alpha = \alpha + \frac{\sum_{i=1}^{n} Z_i \varepsilon_i}{\sum_{i=1}^{n} Z_i^2}.$$

Hence, using Slutsky's theorem, we compute that

$$\mathbb{E}[(r_i^X)^2 | Z] = \frac{Z_i^2}{\sum_{j=1}^{n} Z_j^2} \left( \frac{\sum_{j=1}^{n} Z_j^2 \sigma_{X,j}^2}{\sum_{j=1}^{n} Z_j^2} + \sigma_{X,i}^2 \right) + \sigma_{X,i}^2$$

which converges in probability as $\frac{1}{n}(C \cdot Z_i^2 + \sigma_{X,i}^2) + \sigma_{X,i}^2$ where $C$ is a constant. Recall the definition of the set of nearest neighbours $\mathcal{S}_i$ from assumption (A5). Following the results in Hall and Carroll [1989], we establish the convergence of $\hat\sigma_{X,i}^2 = \frac{1}{\lambda} \sum_{r_j \in \mathcal{S}_i} r_j^2$. Furthermore, we see that the smoothed variances of the residuals behave asymptotically as

$$\hat\sigma_{X,i}^2 = \frac{1}{\lambda} \sum_{r_j \in \mathcal{S}_i} r_j^2 \approx \frac{1}{\lambda} \sum_{r_j \in \mathcal{S}_i} \mathbb{E}[r_j^2 | Z]$$

$$= \frac{1}{\lambda} \sum_{r_j \in \mathcal{S}_i} \left( \frac{C \cdot Z_j^2 + \sigma_{X,j}^2}{n} + \sigma_{X,j}^2 \right)$$

$$\approx \sigma_{X,i}^2.$$

After estimation of the variances, we determine the residuals of the WLS regression with weight matrix $\hat{W}_X$ which is equivalent to the OLS regression after scaling all variables by $\hat{W}_X$. In other words, we define

$$R_i^X = (\alpha - \tilde{\alpha})\hat{W}_X Z_i + \hat{W}_X \varepsilon_i,$$

where $\tilde{\alpha}$ is the OLS-estimator w.r.t. the regressor $\hat{W}_X Z$ and the error variable $\hat{W}_X \tilde{\varepsilon}$. Define $\hat{W}_Y$ and $R_i^Y$ analogously.

The following asymptotic moment formulas can be computed from the definition of $R_i^X, R_i^Y$.

**Lemma 3.** *Assume (A1)-(A5). Then* $\mathbb{E}[R_i^X | Z] = 0$ *and for large* $n$

$$\mathbb{E}[\left(R_i^X\right)^2 | Z] \approx 1 + \frac{Z_i^2 + 2Z_i}{\sum_{j=1}^n Z_j^2}$$

$$\mathbb{E}[\left(R_i^X\right)^4 | Z] \approx 3 + \frac{p(Z_i)}{\left(\sum_{j=1}^n Z_j^2\right)^4},$$

*where* $p$ *is a polynomial of degree* $4$. *In particular, by (A2),(A5) and the fact that for Gaussians higher moments are functions of the variance,* $\mathrm{Var}(\left(R_i^X\right)^2) \leq K$ *for some constant* $K > 0$ *independent of* $i$. *The same statements hold for the residuals* $R_i^Y$.

**Lemma 4.** *Assume (A1)-(A5). Then, the estimator*

$$\hat{\rho}_{X,Y|Z} = \rho(R^X, R^Y) = \frac{\frac{1}{n}\sum_{i=1}^n R_i^X R_i^Y}{\sqrt{\frac{1}{n}\sum_{i=1}^n (R_i^X)^2}\sqrt{\frac{1}{n}\sum_{i=1}^n (R_i^Y)^2}}$$

*consistently estimates* $\rho_{X,Y|Z}$.

*Proof.* By Slutsky's theorem, it suffices to prove that the three sums in the definition of $\hat{\rho}_{X,Y|Z}$ converge in probability and to compute the limits. Convergence follows from the law of large numbers and the variance bounds $\mathrm{Var}(\left(R_i^X\right)^2) \leq K, \mathrm{Var}(\left(R_i^Y\right)^2) \leq K'$ of Lemma 3. The limit of the numerator is equal to $\lim_{n\to\infty} \frac{1}{n}\sum_{i=1}^n E[R_i^X R_i^Y]$, so we compute this quantity first conditioned on $Z$. This yields for large $n$

$$\frac{1}{n}\sum_{i=1}^n \mathbb{E}[R_i^X R_i^Y | Z] \approx \frac{1}{n}\sum_{i=1}^n \rho_{X,Y|Z}\left(1 + \frac{3Z_i^2}{\sum_{j=1}^n Z_j^2}\right) \approx \rho_{X,Y|Z}.$$

Hence $\lim_{n\to\infty} \frac{1}{n}\sum_{i=1}^n \mathbb{E}[R_i^X R_i^Y] = \rho_{X,Y|Z}$. To compute $\lim_{n\to\infty} \frac{1}{n}\sum_{i=1}^n \mathbb{E}[(R_i^X)^2]$ we use the first formula of Lemma 3 to see that

$$\frac{1}{n}\sum_{i=1}^n \mathbb{E}[(R_i^X)^2 | Z] \approx 1 + \frac{1}{n}\sum_{i=1}^n \left(\frac{Z_i^2 + 2Z_i}{\sum_{j=1}^n Z_j^2}\right) = 1 + \frac{1}{n}\left(1 + \frac{\frac{2}{n}\sum_{i=1}^n Z_i}{\frac{1}{n}\sum_{j=1}^n Z_j^2}\right) \approx 1,$$

so that $\lim_{n\to\infty} \frac{1}{n}\sum_{i=1}^n \mathbb{E}[(R_i^X)^2] = 1$.

A similar computation yields $\lim_{n\to\infty} \frac{1}{n}\sum_{i=1}^n \mathbb{E}[(R_i^Y)^2] = 1$, so that indeed $\hat{\rho}_{X,Y|Z} \to \rho_{X,Y|Z}$ as $n \to \infty$ in probability. $\square$

### A.3 Proof of Theorem 2

*Proof.* Assumption 5 allows us to apply ParCorr-WLS in every conditional independence test within the PC algorithm. Using Theorem 1, we get the result. $\square$

### A.4 Extension of Theorem 2 to different heteroskedasticity forms

As discussed in section 4.2, the PC algorithm with ParCorr-OLS is not consistent. If we use ParCorr-WLS with a consistent estimate for the variance of the residual in every conditional independence test, we can make the PC algorithm consistent.

The scope of the current work is a situation where heteroskedastic noise can be dependent on one of the parents of the node or on the sampling index. However, only for the stricter set of assumptions for Theorem 2 we can prove consistency, namely only for sample-index dependent heteroskedasticity.

In the following, we will discuss different situations of heteroskedasticity affecting variables involved in the CI tests within the PC algorithm. In some of these situations, we are able to apply our proposed ParCorr-WLS CI test, thus obtaining consistency. However, in other situations we are not able to use background knowledge about heteroskedasticity and therefore suffer the same problems as in the standard version of the PC algorithm with ParCorr-OLS.

Let us look at the following cases.

**Case 1: The heteroskedasticity in $X$ is independent of that in $Y$.** In case of direct heteroskedasticity, we apply ParCorr-WLS. However, if we encounter a situation where the heteroskedasticity in either $X$ or $Y$ is caused indirectly through a parent, compare figure 4, we lack the background knowledge about it, and therefore apply ParCorr-OLS. Due to Effect 1, we get type I error control for both ParCorr-OLS and ParCorr-WLS. With regard to type II errors, by Effect 2 it is clear that the power of ParCorr-OLS is reduced.

In a setting where the sample size is large enough to get a reasonable weight approximation for WLS, we can only gain detection power by using ParCorr-WLS within the PC algorithm.

**Case 2: There is dependence between the heteroskedasticity in $X$ and that in $Y$.** If $X$ and $Y$ are uncorrelated, i.e. $\rho_{X,Y|Z} = 0$, and the heteroskedasticity of at least one of them is directly induced by a parent or by the sampling index, then we apply WLS whereby we remove the heteroskedasticity and are in Case 1 or in the homoskedastic case again.

If the heteroskedasticity is indirect, meaning one of the parents of $X$ and/or $Y$, that is not included in the conditioning set, is heteroskedastic, our expert knowledge doesn't include this information and we perform ParCorr-OLS. By Effect 1, we know that this suffers from inflated false positives. However, if the link is not removed in this step of the PC algorithm, the conditioning set will be increased in subsequent steps. This means at some point it will include the parent that caused the indirect heteroskedasticity, if the link did not get removed before. When this happens we again will be in Case 1 or in the homoskedastic case.

The main challenge, or in other words the reason why we cannot prove consistency for this more general form of heteroskedasticity, arises if $X$ and $Y$ are correlated. If at least one of them is affected by direct heteroskedasticity, then we again apply WLS whereby we remove the heteroskedasticity and are in Case 1 or in the homoskedastic case as above.

But, if the heteroskedasticity is indirect in both $X$ and $Y$, then we apply ParCorr-OLS. Since the power for this test is very low in this situation, and we do not obtain a consistent estimator for the standard deviation of the regression parameter, we potentially remove the link between $X$ and $Y$.

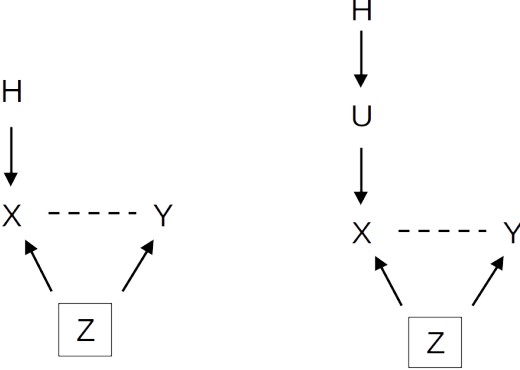

Figure 4: Illustration of Case 1. The dashed line indicates the relationship between $X$ and $Y$ which we are testing. On the left, only $X$ is affected by direct heteroskedasticity introduced by $H$. In this situation, we would apply ParCorr-WLS since the information about this heteroskedasticity is included in the expert knowledge. On the other hand, in case of indirect heteroskedasticity (right), we do not know about the heteroskedasticity in $X$ because we do not know whether $U$ causes $X$ before running the causal discovery algorithm. Therefore, we would apply ParCorr-OLS, which potentially leads to reduced power.

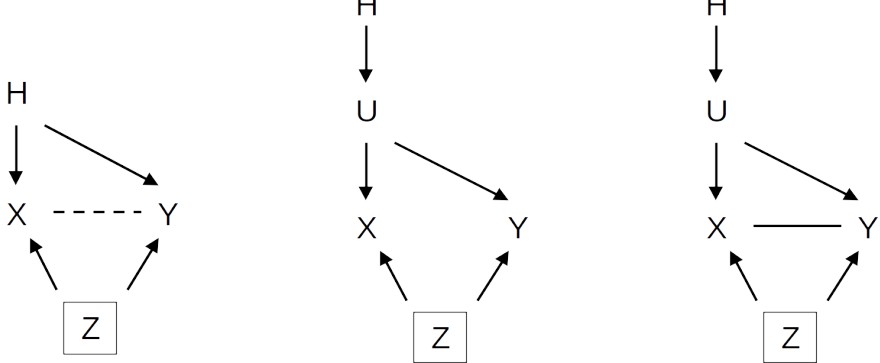

Figure 5: Illustration of Case 2. Here $X$ and $Y$ are both affected by the same kind of (indirect) heteroskedasticity. On the left, the heteroskedasticity is directly introduced by parent $H$. In this situation, we would apply ParCorr-WLS since the information about this heteroskedasticity is included in the expert knowledge. On the other hand, in case of indirect heteroskedasticity (middle and right), we do not know about the heteroskedasticity in $X$ and $Y$ because we do not know whether $U$ causes $X$ and/or $Y$ beforehand. Therefore, we would apply ParCorr-OLS. If there is no ground truth link between $X$ and $Y$ (middle), this could lead to the wrong detection of a link. However, this means that we would enlarge the conditioning set by including $U$ in a later PC algorithm step. Thereby, the heteroskedasticity is removed. If $X$ and $Y$ are dependent as indicated by the solid line (right), applying ParCorr-OLS might lead to missing this link, i.e. to inconsistency in the PC algorithm.

## A.5    Additional Plots

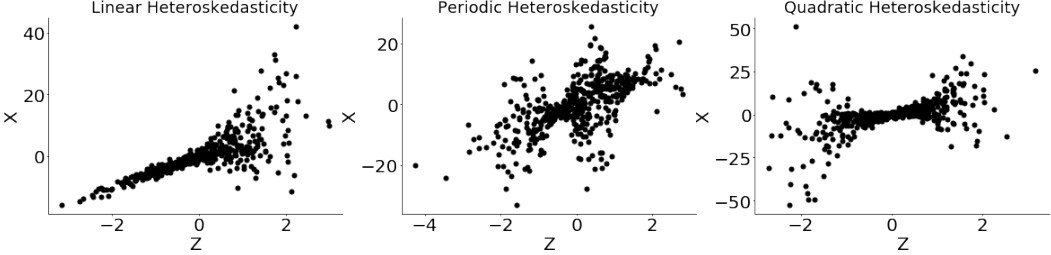

Figure 6: Visualizations of the skedasticity functions $h$ that depend on the confounder $Z$ for strength $s = 5$, compare equation 3.

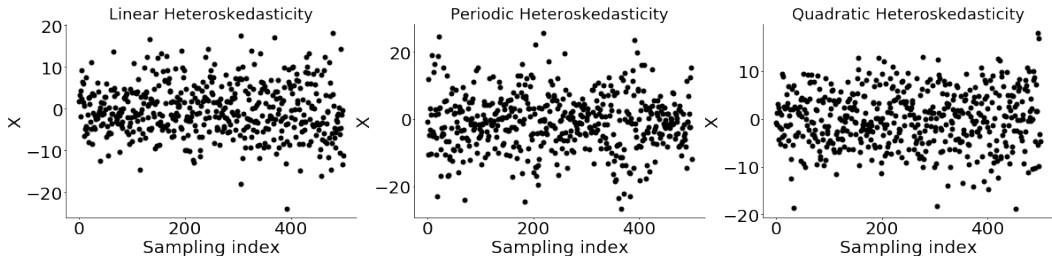

Figure 7: Visualizations of the skedasticity functions $h$ that depend on the sampling index for strength $s = 5$, compare equation 3.

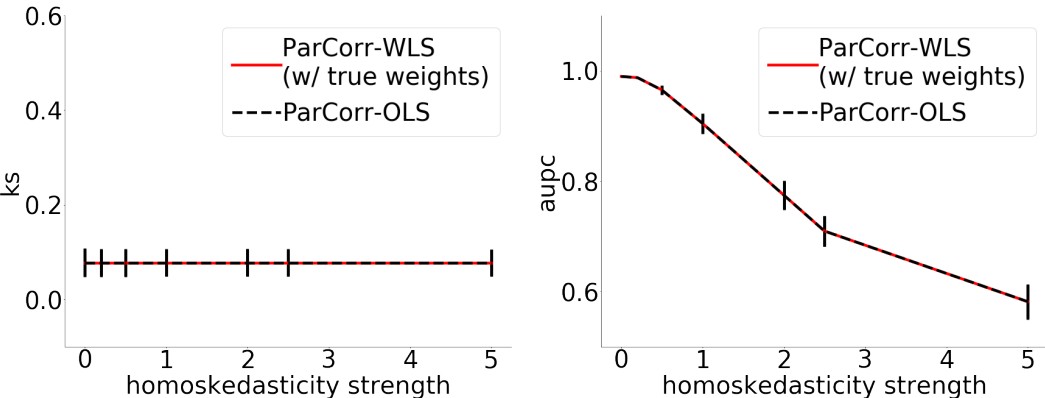

Figure 8: Performance of partial correlation CI tests for dependence ($c = 0.5$) and conditional independence ($c = 0$) between $X$ and $Y$ given $Z$. Shown are KS (left) and AUPC (right) for different strengths $s$ of noise but the noise is homoskedastic. In other words, we consider the skedasticity function $h_X \equiv s$. This figure illustrates that the power of ParCorr-WLS and ParCorr-OLS reduces in the same way on increasingly noisy but homoskedastic data.

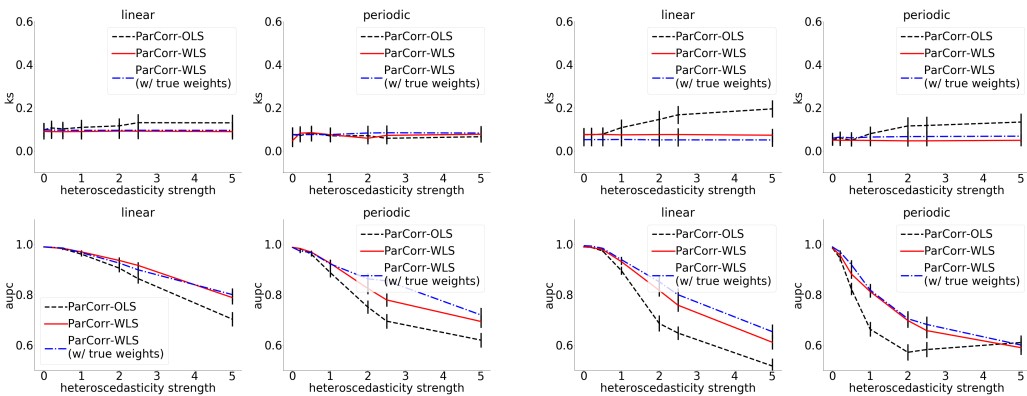

Figure 9: Performance of partial correlation CI tests for dependence ($c = 0.5$) and conditional independence ($c = 0$) between $X$ and $Y$ given $Z$. In all plots the heteroskedasticity is a function of the sampling index and only affects $X$ (left two columns) or both $X$ and $Y$ (right two columns). Shown are KS (top row) and AUPC (bottom row) for different strengths of heteroskedasticity for linear (left), periodic (right) noise scaling functions. The ground truth weights are used for WLS, or the weights are estimated using the window approach with window length 10 as detailed in section 4.1. A sample size of 500 is used and the experiments are repeated 100 times.

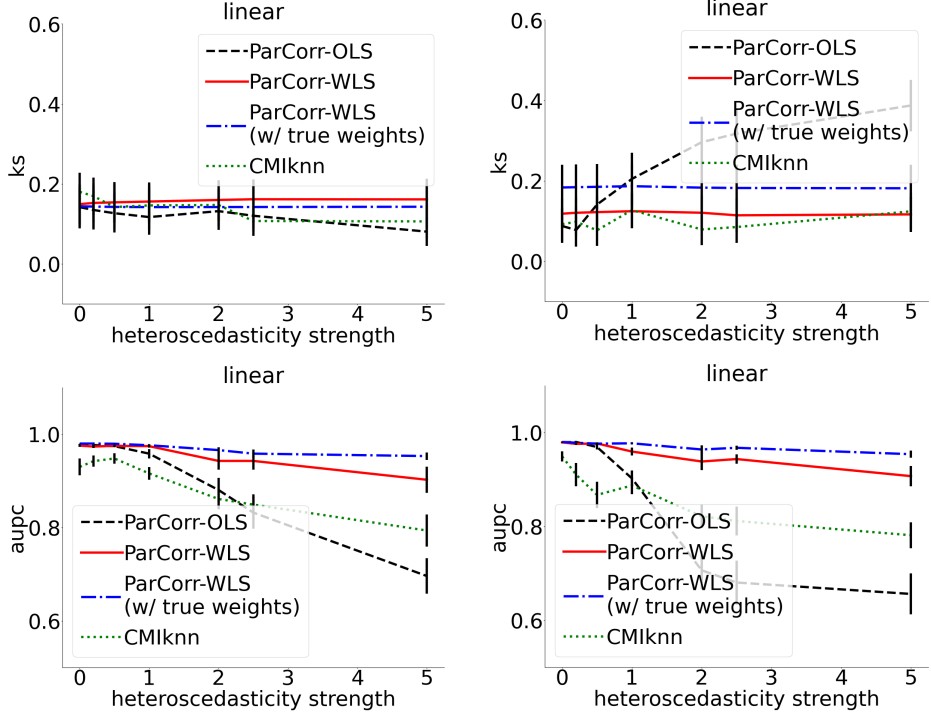

Figure 10: Performance of partial correlation CI tests for dependence ($c = 0.5$) and conditional independence ($c = 0$) between $X$ and $Y$ given $Z$. We compare our method to ParCorr-OLS and the non-parametric CI test CMIknn [Runge, 2018]. In all plots the heteroskedasticity is a function of $Z$ and only affects $X$ (left two columns) or both $X$ and $Y$ (right two columns). Shown are KS (top row) and AUPC (bottom row) for different strengths of heteroskedasticity for linear noise scaling functions. The ground truth weights are used for WLS, or the weights are estimated using the window approach with window length 10 as detailed in section 4.1. A sample size of 500 is used and the experiments are repeated 50 times. For CMIknn we use 0.1 as the number of nearest-neighbors around each sample point, otherwise the default values are used.

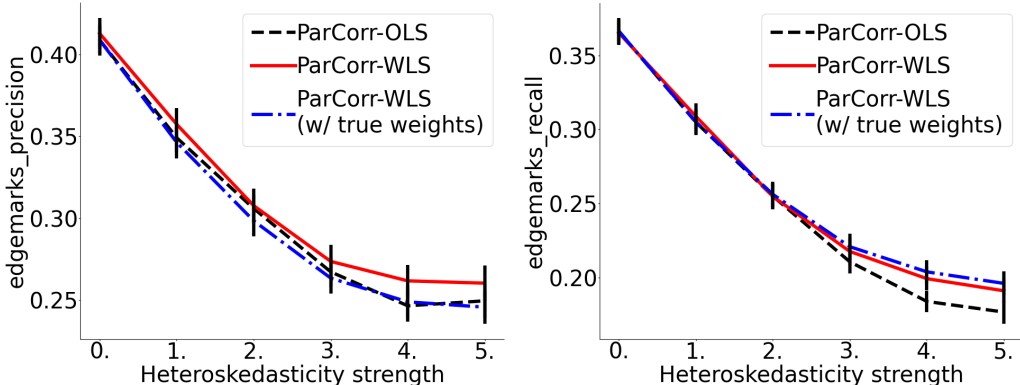

Figure 11: Results for the PC algorithm with the standard ParCorr-OLS CI test compared to that with the proposed ParCorr-WLS test. Shown are edgemark precision ( right) and recall (left) for increasing strengths of heteroskedasticity. The graph has 10 nodes and 10 edges, a sample size of 500 is used. The significance level $\alpha$ is set to 0.05. The experiment is repeated 500 times. Errorbars show standard errors. Estimated weights with a window length of 5 or ground truth weights are used for WLS.