# OpenReview forum: "Conditional Independence Testing with Heteroskedastic Data and Applications to Causal Discovery"
_NeurIPS.cc/2022/Conference — NeurIPS 2022 Accept_

### Official Review · Reviewer_9b5D · 2022-07-11

**Rating:** 6
**Confidence:** 4
**Soundness:** 3 good
**Presentation:** 2 fair
**Contribution:** 2 fair

**Summary:**

In this work, authors consider heteroskedasticity in a structural causal model (SCM) and develop an adapted weighted least-squares (WLS) partial correlation variant as a conditional independence (CI) test which is able to deal with heteroskedastic noise with expert knowledge about the heteroskedastic relationships are given. Combined with the PC algorithm, the proposed CI test can also identify the causal relationships between variables under some assumptions regarding the heteroskedastic relationships. Both theoretical consistency analysis and simulation experiments demonstrate that the proposed CI test achieves better false positive rates and also improves upon the detection power as compared to the standard partial correlation CI test.

**Questions:**

1. The authors did not mention the computational complexity of the proposed causal discovery algorithm (i.e., inject the proposed CI test into the PC algorithm), and it would be great if the authors could discuss the computational complexity of the proposed algorithm.

2. Since only simulated experiments are conducted in the paper, the reviewer wonders whether the proposed algorithm works well on real-world datasets?

**Limitations:**

Authors have discussed the Limitations and societal impacts in the Conclusion Section.


**Strengths And Weaknesses:**

Strengths:

1. The authors propose an adapted WLS partial correlation test for conditional independence testing in the presence of heteroskedastic noise.

2. The authors provide a detailed analysis of the WLS method's consistency with the proposed weight approximation method, and further inject the proposed CI test into the PC algorithm to learn the causal structure.

3. Experiments on simulated datasets demonstrate that the proposed WLS partial correlation CI test achieves better performance than the OLS partial correlation CI test in the presence of heteroskedastic noise. Combined with the PC algorithm, the proposed WLS partial correlation CI test has a smaller false positive rate in identifying causal relationships.

Weaknesses:

1. The idea of using the inverse of error's variance is intuitive, however, it would be great if the authors could further discuss why WLS can deal with heteroskedastic noise and some possible better re-weighting strategies in detail.

2. The authors formulate several assumptions in the paper, however, the heteroskedastic relationships require expert knowledge. The reviewer wonders if the authors could discuss further how to relax the heteroskedastic relationships assumption.

---

> ### Author Response · Authors · 2022-08-02
> **Official Response to Reviewer 9b5D**
>
> We thank the reviewer for the positive comments and valuable feedback, specifically for recognizing the detailed analysis of the consistency of our method.
>
> We comment on the raised questions in detail below.
>
> **Response to Weakness 1:** Our Lemma 1 states: if the weights are chosen as the reciprocal of the error variance per sample, the WLS estimator is consistent, efficient, and asymptotically normal, as well as BLUE. This is the case since this transformation of the data removes the heteroskedasticity by standardizing the error variance. Therefore, if the ground truth variance of the errors is known, using its inverse is the best strategy for dealing with heteroskedastic data. Please also refer to lines 167 f. in the paper.
>
> **Response to Weakness 2:** Regarding **Assumption 1 ("Heteroskedastic relationships'')**, it is possible to adapt the variance function estimation step to the multi-dimensional setting [1], i.e. dependence of the noise scaling function on multiple parents, or on a subset of the parents as well as the sampling index. Please also refer to the discussion in Section 6, specifically lines 327-330.
>
> To relax the expert knowledge part of Assumption 1, i.e. knowledge about what the noise scaling function depends on, it would be necessary to adapt the smoothing step of the variance function estimation. Since in this case we would not be able to average residuals that are close in value of the heteroskedasticity-inducing entity, we would need to include all parents and the sampling index to judge closeness of the residuals. This approach would also require us to use an iterative causal discovery scheme to obtain preliminary knowledge about the causal relationships, which comes with additional challenges such as selective inference and computational burden.
>
> **Assumption 5 ("Heteroskedastic relationships regarding PC algorithm'')** is needed to prove consistency of the PC algorithm with our proposed ParCorr-WLS CI test since we need to account for indirect heteroskedasticity in this situation. By indirect heteroskedasticity, we mean heteroskedasticity that is due to a heteroskedastic parent that is not part of the conditioning set, i.e. the heteroskedasticity originates from an ancestor that is not a parent. This kind of heteroskedasticity is not part of the expert knowledge. We discussed this issue further in Section A.4 in the supplement. Note that, empirically, violating this assumption and just fulfilling Assumption 1 still leads to improved performance compared to ParCorr-OLS as is shown by our experiments, in particular Figure 3.
>
> **Answer to Question 1:**
> ParCorr-WLS with true weights, i.e. when the error variance is known, has the same complexity as ParCorr-OLS. The weight approximation step has the following complexity, where $n$ denotes the sample size and $d$ the number of variables in the conditioning set:
> - two linear regressions: $\mathcal{O}( n d^2 + d^3)$
> - sorting by value of the heteroskedasticity-introducing parent value: $\mathcal{O}( n \log (n))$
>
> Therefore, we obtain a complexity of $\mathcal{O}( n d^2 + d^3 + n \log (n))$ which is introduced by the weight approximation step of ParCorr-WLS as compared to $\mathcal{O}( n d^2 + d^3)$ for standard ParCorr.
>
> **Answer to Question 2:** Our method rests on assumptions and expert knowledge that is difficult to obtain without domain expertise. Our experiments on simulated data suggest that the method performs well whenever such knowledge is available. Therefore, we regard it as a step towards improved performance of conditional independence testing and constraint-based causal discovery on real world data.
> We aim to evaluate our method in collaboration with domain experts in the future.
>
>
> We thank the reviewer again for the feedback which helps to improve the paper. We hope that addressing your concerns raises our work from a fair to a good contribution.
>
> [1] Spokoiny, V. (2002). Variance estimation for high-dimensional regression models. Journal of Multivariate Analysis, 82(1), 111-133.

---

### Official Review · Reviewer_hnmZ · 2022-07-12

**Rating:** 7
**Confidence:** 4
**Soundness:** 4 excellent
**Presentation:** 3 good
**Contribution:** 3 good

**Summary:**

A conditional independence test is proposed based on partial correlation, but adapted to the case of heteroskedasticity. It is also studied how this test can be used in the PC algorithm. All results a rigorously proved, as well as illustrated using experiments.

**Questions:**

* On line 102, I was confused by the $[0, \infty)$. This seems to be the domain of $t$, though as far as I can tell, this is always a discrete sampling index, not a continuous value.

* On line 270-271, it is described that the case of "parent and child being affected by the same kind of heteroskedasticity" is being avoided.  I suppose this is simply because that could lead to much larger heteroskedasticity strengths than what is supposed to happen for a given value of the parameter $s$. Or is there a  more fundamental problem with this case?

Minor:

* Line 122-123: "the" missing before "hat operator"

**Limitations:**

Yes

**Strengths And Weaknesses:**

This contribution is of high theoretical quality. It is also of great practical relevance, as homoskedasticity assumptions are often violated in real datasets. There are currently some major limitations (which are clearly mentioned in the paper), mainly that the proposed partial correlation test only works if for each variable, there is at most one source of heteroskedasticity, and for theoretical consistency of the PC algorithm, that this source is not another variable. Improvement in these areas is listed as future work.

The paper is clearly written; I especially appreciate the illustrations of the problems in Figure 1. The application to causal discovery is original, though if I understand correctly, the test itself is quite similar to existing work on heteroskedasticity.

---

> ### Author Response · Authors · 2022-08-02
> **Official Response to Reviewer hnmZ**
>
> We thank the reviewer for the positive comments and valuable feedback, specifically for recognizing the high theoretical quality, practical relevance, and clear presentation of our work.
>
> One of the reviewer's concerns seems to be the similarity to existing work. However, we believe the application to causal discovery is relevant and novel. The theoretical part of our paper, specifically the discussion of consistency of the PC algorithm with our ParCorr-WLS CI test under heteroskedasticity, also adds to our contribution.
>
> We comment on the raised questions in detail below.
>
>
> **Answer to Question 1:** Thank you, indeed in practical situations the sampling index (e.g. time) is discrete. We rephrased the problem statement to include the discrete sampling index by introducing an index set $\mathcal{T}$ such that $t \in \mathcal{T}$.
>
> **Answer to Question 2:** Consider the following model
> $$
> X = a Z + N_X,
> $$
> where $a$ is a constant, $X$, $Z$ and $N_X$ are random variables.
> If parent $Z$ and child $X$ are affected by the same kind of heteroskedasticity, and we are conditioning on the parent within the CI test, the noise term $N_X$ will not be heteroskedastic anymore since the heteroskedasticity is completely explained by the parent. This is then a homoskedastic problem, and we would not be able to highlight the strengths of ParCorr-WLS in such a situation.
>
> **Typos:** We have fixed the typos in the paper, thanks.
>
> We thank the reviewer again for the feedback which makes this a better paper. We hope that addressing your concerns inclines you to increase your rating of our work.

---

### Official Review · Reviewer_vG5d · 2022-07-13

**Rating:** 6
**Confidence:** 3
**Soundness:** 3 good
**Presentation:** 3 good
**Contribution:** 2 fair

**Summary:**

This paper proposes a conditional independence test in a linear Gaussian setting but the variance of the error term may not be a constant but affected by a function of a subset of its parents or sampling index (e.g., individual-dependent variance or a time-series like setting). Hence, the authors designed heteroskedasticity-aware weighted partial correlation where the weights come from inverse of variances, which is estimated using a windowing approach. However, this test works with expert knowledge about heteroskedasticity (whether it is caused by sampling index or heteroskedastic parent). In a causal discovery setting, not knowing the parents of variables, it only assumes that the heteroskedasticy comes from sampling index.

**Questions:**


- What if we can use non-parametric conditional independence test? Will it be (certainly) even weaker than OLS?
- Can we combine both heteroskedastic parent and sampling index? In such a case, can we just use “sampling index” approach? (or impossible to combine them?)
- In estimating weights, no clear justification is given for sorting residuals with respect to the values of H. I understand that being closer in H values implies having similar variances. But I would like to see more formal statements on this.
- I am not sure whether a similar idea is employed in a similar or different setting. Like, transforming data in a way to be more homoskedastic, and then applying ParCorr-OLS.


**Ethics Review Area:**

["I don’t know"]

**Limitations:**


- Requires prior knowledge on the reasons for heteroskedasticity.


**Strengths And Weaknesses:**


Strengths
- well-motivated problem and a theoretically correct solution (under some assumptions.)
- a simple solution to an important problem.

Weaknesses

- As the authors acknowledged, Assumption 1 (and 5 for PC algorithm) requires expert knowledge.

It seems hard for me to find weaknesses of the paper considering that the authors did a right thing to solve a problem and acknowledged limitations clearly that are not due to the authors but comes from the nature.

---

> ### Author Response · Authors · 2022-08-02
> **Official Response to Reviewer vG5d**
>
> We thank the reviewer for the positive comments and valuable feedback, specifically for recognizing the good motivation and theoretical correctness of our solution.
>
> One of the reviewer's concerns seems to be the similarity to existing work. However, we believe the application to causal discovery is interesting and novel. The theoretical part of our paper, specifically the discussion of consistency of the PC algorithm with our ParCorr-WLS CI test under heteroskedasticity, also is a valuable contribution. Another point raised by the reviewer is that the assumptions, on which our method rest, require expert knowledge, and we focus on the one-dimensional case.
> We explored the problem with theoretical rigour given these assumptions and discuss certain caveats that arise with their possible relaxations in the supplementary material. Due to space constraints and an already extensive supplement we point to relevant literature for a multi-dimensional extension.
>
> We comment on the raised questions in detail below.
>
> **Answer to Question 1:** Thanks for the question. We have added experiments that compare our method to the non-parametric CI test CMIknn suggested in [1]. We chose CMIknn because it is suitable for multiplicative confounding as introduced by heteroskedasticity, which not many other CI tests can deal with. Additionally, it has been shown that it has higher power than kernel-based tests in low dimensional problems. The corresponding plots can be found in section A.5. As expected, since the non-parametric test does not rest on assumptions about the functional relationship, the experiments show that the CMIknn CI test has lower power than both ParCorr variants on homoskedastic data, and lower power than ParCorr-WLS as the heteroskedasticity strength increases. The type I error control of the CMIknn CI test is also clearly visible. However, the biggest drawback of CMIknn is its long average runtime.
>
> **Answer to Question 2:** We have focused on heteroskedasticity induced either by one of the parents or by the sampling index in this work to avoid additional challenges that come from approximating the variance function in dimensions larger than one.
> However, it is possible to extend the k-nearest neighbour variance function estimation approach, which we employed, to the multi-dimensional setting, see e.g. [3]. The idea is the same as in the univariate case. Except that we need to smooth over residuals that are close in H values as well as in sampling index.
>
> **Answer to Question 3:** We base our variance approximation scheme on the work done by Robinson [2], see pages 878 f. for a more formal description of the estimator.
>
> We restate the idea for the case where $H$ is part of the conditioning set in the following:
> Consider again the SCM (2) where we focus on the case of $H$-dependent heteroskedasticity, i.e.
> $
> X_i = aH_i + \sigma^2(H_i) · N_X, ~ i=1,\ldots, n.
> $
> The goal is to approximate $\sigma^2(h) = \operatorname{Var}(X|H = h)$. Since
> $
> \operatorname{Var}(X|H = h) = \mathbb{E}[(X - \mathbb{E}[X | H=h])^2 | H=h],
> $
> this is the regression of $(X-aH)^2$ on $H$. In other words, to estimate $\sigma^2(H_i)$ we need to regress $(X_i-aH_i)^2$ on $H_i$. This motivates the estimation method of first using OLS regression to calculate the residuals $r_i = X_i-\hat{a}H_i$ and then using a non-parametric regression method to regress these residuals on $H$. As suggested in [2] we use a linear combination of the $r_j^2$ corresponding to the $k$ closest $H_j$ to $H_i$. We find the $k$ closest residuals in $H$-value by sorting w.r.t. $H$.
>
> **Answer to Question 4:** In many application settings WLS is used to deal with heteroskedastic data, e.g. [4] make the case for combining WLS with heterokedasticty-consistent standard errors in econometrics, and [5] describes variants of WLS for dealing with heteroskedasticity in the context of medicine.
> To the best of our knowledge, the application to CI testing and (subsequently) causal discovery is novel. We also believe that our theoretical treatment of the problem adds value.
>
> We thank the reviewer again for the helpful feedback. We hope that addressing your concerns raises our work from a fair to a good contribution.
>
>
> [1] Runge, J. (2018). Conditional independence testing based on a nearest-neighbor estimator of conditional mutual information. In International Conference on Artificial Intelligence and Statistics (pp. 938-947). PMLR.
>
> [2] Robinson, P. M. (1987). Asymptotically efficient estimation in the presence of heteroskedasticity of unknown form. Econometrica: Journal of the Econometric Society, 875-891.
>
> [3] Spokoiny, V. (2002). Variance estimation for high-dimensional regression models. Journal of Multivariate Analysis, 82(1), 111-133.
>
> [4] Romano, J. P., & Wolf, M. (2017). Resurrecting weighted least squares. Journal of Econometrics, 197(1), 1-19.
>
> [5] Beal, S. L., & Sheiner, L. B. (1988). Heteroscedastic nonlinear regression. Technometrics, 30(3), 327-338.

---

> > ### Comment · Reviewer_vG5d · 2022-08-08
> > **follow-up**
> >
> > Thanks for a detailed reply with new experimental results. I am keeping my positive assessment of the paper (together with other reviewers.)

---

### Comment · Area_Chair_g9cy · 2022-08-08
**Please respond to author response**

Dear reviewers,

Thank you for reviewing this paper. Could you respond to the author feedback, or at least acknowledge that you have read the response (thanks to reviewer vG5d for having done it)? Please indicate whether the author response addresses your concerns.

Thanks,
AC

---

### Meta-Review · Area_Chair_g9cy · 2022-08-28

**Recommendation:** Accept
**Confidence:** Less certain

**Metareview:**

In real problems, data often exhibit the heterogeneity property. As a consequence, conditional independence tests that rely on the data homoskedasticity assumption may perform suboptimally, leading to inaccurate causal discovery results. This paper adapts the partial correlation tests to account for heteroskedastic noise and provides some necessary theoretical guarantees and empirical results. Reviewers agree that the studied problem is well motivated and that the solution is sensible.

**Award:**

No

---

### Decision · Program_Chairs · 2022-09-14

Accept